# Chaperones—A New Class of Potential Therapeutic Targets in Alzheimer’s Disease

**DOI:** 10.3390/ijms25063401

**Published:** 2024-03-17

**Authors:** Joanna Batko, Katarzyna Antosz, Weronika Miśków, Magdalena Pszczołowska, Kamil Walczak, Jerzy Leszek

**Affiliations:** 1Faculty of Medicine, Wroclaw Medical University, Ludwika Pasteura 1, 50-367 Wrocław, Poland; joanna.batko@student.umw.edu.pl (J.B.); katarzyna.antosz@student.umw.edu.pl (K.A.); miskowweronika0@gmail.com (W.M.); magdalena.pszczolowska@gmail.com (M.P.);; 2Clinic of Psychiatry, Department of Psychiatry, Wroclaw Medical University, Ludwika Pasteura 10, 50-367 Wrocław, Poland

**Keywords:** Alzheimer’s disease, dementia, chaperones, Hsp90, Hsp60, Hsp70, clusterin

## Abstract

The review describes correlations between impaired functioning of chaperones and co-chaperones in Alzheimer’s disease (AD) pathogenesis. The study aims to highlight significant lines of research in this field. Chaperones like Hsp90 or Hsp70 are critical agents in regulating cell homeostasis. Due to some conditions, like aging, their activity is damaged, resulting in β-amyloid and tau aggregation. This leads to the development of neurocognitive impairment. Dysregulation of co-chaperones is one of the causes of this condition. Disorders in the functioning of molecules like PP5, Cdc37, CacyBP/SIPTRAP1, CHIP protein, FKBP52, or STIP1 play a key role in AD pathogenesis. PP5, Cdc37, CacyBP/SIPTRAP1, and FKBP52 are Hsp90 co-chaperones. CHIP protein is a co-chaperone that switches Hsp70/Hsp90 complexes, and STIP1 binds to Hsp70. Recognition of precise processes allows for the invention of effective treatment methods. Potential drugs may either reduce tau levels or inhibit tau accumulation and aggregation. Some substances neuroprotect from Aβ toxicity. Further studies on chaperones and co-chaperones are required to understand the fundamental tenets of this topic more entirely and improve the prevention and treatment of AD.

## 1. Introduction

It is estimated that the number of patients who have Alzheimer’s disease (AD) will be growing significantly. In the USA, currently, there are 6 million people affected by this condition, and their number is expected to increase to 13 million in 2050 [1]. This disease, due to the gradual limitation of patients’ independence in everyday activities, generates enormous costs—unpaid guardians like closest family and friends provided 18 billion hours of guardianship valued at $339.5 billion. The Alzheimer Association estimates that 1.2 additional job positions will be needed to care for the growing AD patient population. In 2050, total payments for the care of people living with dementia are predicted to be nearly $1 trillion [2].

This is why the pathophysiology of AD is analyzed in detail, with a concentration on events at the cell level. Scientists believe that focusing on the condition’s molecular pattern is necessary to create a successful therapy scheme. Recent discoveries point to the significant impact of the dysfunction of chosen proteins, chaperones, and co-chaperones on AD pathogenesis [3]. Molecules like chaperones were discovered nearly 50 years ago. Their main aim is to preserve the proteasome in its correct shape under various conditions [4]. Many studies that tried to explain its functioning have been conducted all over the years; however, there are still many questions regarding those molecules, like their specificity for client proteins or the possibility of using them in the therapy of various diseases. In this comprehensive narrative review, these correlations are investigated and described.

## 2. Alzheimer’s Disease

Alzheimer’s disease (AD) is a neurodegenerative disease and the most common cause of dementia; it accounts for 60 to 80% of cases [4]. It is caused by the accumulation of amyloid-beta (Aβ) outside of neurons, which interferes with transmission between synapses and hyperphosphorylated tau proteins inside the cells, blocking the transportation of nutrients and molecules that nourish neurons. Also, cholinergic neurons in the nucleus basalis of Meynert undergo degeneration, which causes dysfunction in the cholinergic system. Aβ deposits activate proinflammatory cytokines and cause chronic inflammation that disrupts microglial clearance of Aβ [5]. 

The risk factors for AD are, among others, age, genetics (APOE-e4), Down syndrome, genetic mutations involving the APP gene and genes for presenilin-1 and presenilin-2, first-degree relative with AD, smoking, midlife obesity, hypertension, high cholesterol levels, diabetes, diet, low physical activity, lower socioeconomic status, and poor sleep quality. The existence of protective factors has also been demonstrated, including physical activity, a heart-healthy diet such as Mediterranean Dietary Approaches to Stop Hypertension (DASH) or Mediterranean-DASH Intervention for Neurodegenerative Delay (MIND), multiple daily activities, more years of formal education, and remaining socially and mentally active throughout life [4].

Some of the main signs of AD are memory loss that disrupts daily life, problems in planning and solving problems, poor judgment, confusion with time and place, changes in personality and behavior, for example, depression or apathy, withdrawal from social activities, challenges in writing, speaking, and walking, and dysphagia. Those symptoms progress, and we can distinguish several phases in the course of the disease: preclinical AD, Mild Cognitive Impairment (MCI) due to AD, and dementia due to AD, which increases in severity over time [4].

The diagnosis of AD is made by finding symptoms of dementia and excluding other causes such as vitamin B12 deficiency, hypothyroidism, HIV, or depression. Structural brain imaging (MRI and TK) can show hippocampal and cortical atrophy in the temporal and parietal regions. Fluoro-deoxy-glucose (FDG) and amyloid PET are also used in diagnosis. There are also changes in the cerebrospinal fluid, such as low Aβ, high tau, and phospho-tau levels [6].

Current strategies in the pharmacological treatment of AD involve acetylcholinesterase inhibitors: donepezil, rivastigmine, and galantamine, which improve cognition, and the NMDA receptor antagonist memantine, used in moderate to severe forms of AD [5]. New therapies involving monoclonal antibodies also target Aβ, for example, aducanumab and gantenerumab [7,8]. Many trials are being held to find new targets that can be used in AD treatment, including ones targeting chaperones, which are the main topic of our review. 

## 3. Chaperones

In a living organism, such as a human body built from various proteins, the proper, adequate manner of structural formation seems crucial for the whole organism’s efficiency. Several processes occur one after the other to create a protein molecule that presents a specific conformation and is appropriately prepared to perform the assigned function. Referring to protein conformation, when amino-acid chains are created, there is a need to create the given three-dimensional structure from them. Some of the sequences contain the required information to fold spontaneously. However, not all of them can accomplish the process by themselves [9,10,11].

The chaperones are an enormous facilitation for these biochemical pathways for proper molecule creation. They are a group of specific proteins capable not only of folding the unfolded polypeptides but also of repairing misfolded ones. Additionally, they can prevent protein aggregation as well as direct terminal proteins for proteolytic degradation [9,11]. The amount of misfolded proteins is strongly connected with cellular stresses, which refer to heat shock, oxidative stress, and the vastness of pathological conditions [9] (Figure 1).

The prominent part of chaperone synthesis is induced under conditions of cellular stress (for example, heat shock, oxidative stress, genetic pathology) as an organism’s self-defense due to a destabilization of proteostasis. These chaperones are classified based on their sequence homology, and members of individual groups are named historically according to their molecular weight. Therefore, the Hsp40s, Hsp60s, Hsp70s, Hsp90s, Hsp100s, and the small Hsps can be awarded. Each of these groups has a different function and role in proteostatis [10,12].

The large, barrel-like molecules are called chaperonins, or Hsp60, due to their molecular weight. Chaperonins are part of a much larger family of adenosine triphosphate (ATP)-dependent proteins, which use high-energy (phosphoanhydride) bonds to control the process. Due to their barrel-like shape, chaperonins create perfect conditions inside the “barrel” for polypeptides to rearrange conformation, leading to the native form. Moreover, the process and its requirements protect from aggregation [12,13].

Referring to the ATP-dependent proteins, apart from Hsp60, Hsp70 (ubiquitous family), as well as Hsp90 and Hsp100, present these biochemical properties. Moreover, Hsp70 and Hsp90 are the most studied heat-shock proteins in this very large protein family and will be discussed in more detail in the parts below [14].

Last but not least, small heat shock proteins (Hsps) are separated. The molecular masses of the subunits vary from 12 to 43 kDa. For example, Hsp20, Hsp22, and Hsp25/27 can be mentioned [13]. The small Hsps and Hsp70 act as molecular “clamps” as a form of protection for native protein formation [15] (Table 1).

Additionally, what is worth mentioning is that many chaperones function as oligomers. Their molecular masses, due to this connection, are significantly higher. Regarding a different protein class, for example, Hsp70 operates mainly in tandem with Hsp40 co-chaperones, which affects their molecular masses and final features in vivo [13].

## 4. Chaperonopathies

When taking a closer look, chaperones are the Chaperone (chaperoning) system (CS) elements, which consist of chaperones, co-chaperones, chaperone co-factors, interactors, and receptors. CS is a significant conditioning factor of protein homeostasis (proteostasis) along with the ubiquitin–proteasome system and autophagy [10,16]. Moreover, in this molecular approach, all different kinds of CS defects are the cause of ailments called chaperonopathies. Chaperonopathy is a pathological condition of chaperone structure or function, for example, due to a genetic or acquired defect, which causes CS failure. Therefore, what is worth emphasizing is that from chaperonopathies, neurochaperonopathies can be separated, where diseases such as neurodegenerative and neuromuscular disorders can be found [11]. The imbalance of proteostasis is observed in the formation and accumulation of pathological inclusions, such as α-synuclein in Parkinson’s disease or huntingtin in Huntington’s disease, as well as extracellular β-amyloid plaques in Alzheimer’s disease [10,11,14]. The imbalance of proteostasis is observed as the formation and accumulation of pathological inclusions. Due to the incorrect protein formations, more and more unshapely biological pathways are present, as the proteins are not able to perform their proper function. The accumulation of invalid structures causes vast groups of neurological disorders, such as α-synuclein in Parkinson’s disease or huntingtin in Huntington’s disease, as well as the extracellular β-amyloid plaques in Alzheimer’s disease [10,11,14].

At this point, it is worth asking: What is the flashpoint of chaperonopathies and dysregulations described above? Firstly, the genetic association is a significant one. Unfortunately, there is no direct mutation yet to be indicated precisely, but while comparing early-onset (familial) and late-onset (sporadic), the conclusion could be made that genetically invalid CS is also a possible way of early and rapidly progressing AD [11,17]. 

Secondly, as chaperones are biological structures sensitive to surrounding conditions as well as the rest of the biochemical individuals, the Acquired Neurochaperonopathies develop with time. Some signs of acquired changes in Hsp70 and Hsp90 can cause AD. Changes in enzymes, due to, for example, adenylation (AMPylation), inhibit Hsp70 and therefore lead to decreased cell growth, toxic protein aggregation, and misfolding [11,18]. 

A similar situation happens with phosphorylation of the Cell Division Cycle 37 (Cdc37) that collaborates with Hsp90. With these examples, there is a conclusion that conformation changes, caused by cellular stress, inflammation, and external factors, can lead to CS invalidation [19]. 

The heterogeneity of potential dysregulation factors is enormous, as in any case of protein disfunction. It is important to identify possible points for maneuvers and therapy.

## 5. Clusterin

Clusterin (CLU), or APOJ, is a multifunctional secretory glycoprotein implicated in several physiological and pathological states, including Alzheimer’s disease (AD). CLU can interact with a broad spectrum of molecules. Clusterin activates NF-κB and up-regulates the expression of MMP-9 and TNF-α. TNF-α induced by clusterin was significantly abrogated by pretreatment with TLR4-signaling inhibitors and anti-TLR4 neutralizing antibodies [20]. CLU is mostly synthesized in astrocytes in the brain but is highly inducible in neurons by AD risk factors [21]. Three genetic polymorphisms of clusterin predisposing to cognitive impairment have been detected: rs11136000, rs2279590, and rs9331888 [22]. The CLU allele variant rs11136000 probably predisposes to the development of late-onset Alzheimer’s disease [23]. 

Moreover, elevated clusterin levels correlate with greater severity and faster progression of the disease [24]. CLU functions as an extracellular chaperone and is responsible for lipid transfer and immune modulation. Clusterin is a mainly secreted chaperone found abundantly in plasma and extracellular fluid. During passage through the secretory pathway, immature CLU is extensively N-glycosylated and cleaved into α and β chains, which remain connected by disulfide bonds [25]. However, several studies have demonstrated the presence of intracellular clusterin under stress conditions [26]. CLU also stabilizes unfolded proteins against aggregation and can inhibit the formation of amyloid β (Aβ) fibrils and other amyloidogenic proteins in vitro, consistent with the function of the ATP-independent chaperone “holdase” [25]. Clusterin is involved in pathways common to several diseases, such as oxidative stress, cell death and survival, and proteotoxic stress [27]. In a study in which the cerebrospinal fluid of patients with Alzheimer’s disease (*n* = 99) and control subjects (*n* = 39) was analyzed, patients with AD presented higher clusterin content. Clusterin was quantified both before and after deglycosylation using a sandwich immunoenzymatic assay (ELISA) (before deglycosylation: 7.17 ± 2.43 vs. 5.73 ± 2.09 AU; *p* = 0.002; after deglycosylation: 12.19 ± 5.00 vs. 9.68 ± 4.38 AU; *p* = 0.004 [28]. Clusterin’s ability to interact with and bind to Aβ appears to alter aggregation and promote Aβ clearance, suggesting a neuroprotective role [29]. High plasma clusterin levels in healthy middle-aged adults are associated with reduced volume of the midbrain cortex. This is a brain region that atrophies in early AD. This means that plasma clusterin may serve as a biomarker of preclinical AD. Unfortunately, whether elevated clusterin levels are a consequence of pathology or a promoting factor has not been investigated [30]. 

It has been studied that in patients with various degrees of cognitive impairment, clusterin levels are related to synaptic degeneration and are positively correlated with the level of neurogranin in the cerebrospinal fluid [31]. Another study showed that when injected into the rat hippocampus, clusterin causes increased levels of tau and its phosphorylation. In transgenic mouse models of tau overexpression, clusterin was also overexpressed [32]. The ε4 allele of the apolipoprotein E (APOE4) carriers has a higher percentage of clusterin-containing synapses. In human models of Alzheimer’s disease and mice, APOE4 leads to synaptic degeneration and increases amyloid beta at the synapse in human models of Alzheimer’s disease and mice [33].

Copper limits the amyloidogenic processing of APP [34]. Disturbed copper homeostasis is observed in AD [35]. Copper transport in the brain is regulated by P-type copper transport ATPases (ATP7A and ATP7B) so that levels are sufficient for copper-related proteins but also ensure that toxic excess copper is removed [36]. Clusterin, through the lysosomal pathway, degrades ATP7A and ATP7B [37]. Studies have confirmed the function of clusterin in facilitating autophagy [38] (Figure 2).

## 6. Hsp90, Hsp70, Co-Chaperones

Chaperone proteins such as Hsp90, Hsp70, and co-chaperones play a key role in protein molecular development, influencing their structure and in the regulation of central cellular pathways [39]. Through a well-working complex network of chaperones, cells produce mature proteins that have specific conformations [40]. In the pathological stage, chaperones and co-chaperones fail to fold and degrade by the lysosome; proteins such as tau protein and β-amyloid (main proteins) accumulate in brain tissue in Alzheimer’s disease [41]. The crucial role is to prevent the aggregation of proteins that are not functioning well due to incorrect conformation. The improper structure is a result of a disturbed folding process while the cell is under stress factors like high temperature or chemical factors [42]. This is why the amount of chaperones increases under those circumstances. However, their number decreases with cell aging [43]. 

Hsp90, an essential heat shock protein, accounts for 1–2% of cell proteins, and its amount might double under stress conditions [44]. This protein is responsible for tau, α-synuclein, several kinases, transcription factors, steroid hormone receptors, and E3 ubiquitin ligases appropriate maturation, activation, and degradation, which is crucial for sustaining cell homeostasis [39,42].

Hsp90 consists of three parts: the N-terminal ATP-binding domain, the center domain, and the C-terminal dimerization domain. Hsp90 may be regulated by interactions with co-chaperones, ATP, or heat shock factor (HSF1) [45,46]. In cases of stress conditions, HSF1 disconnects from Hsp90 and connects with heat-shock-factor elements (HSE) on the sequence of Hsp90 encoding gen [47] (Figure 3).

The activity of Hsp90 is also regulated by post-translational modifications (PTMs). Due to PTMs, there are diversifications like recognition, binding, and conformational changes of Hsp90 domains. PTMs like phosphorylation lead to the activation of clients, while hyperphosphorylation inactivates them [48]. Hsp90 may also be regulated by acetylation, S-nitrosylation, or the mediation of Hsp90 ATPase activity. In the last listed process, the important role is played by co-chaperones, proteins that interact with Hsp90 [42]. 

In AD pathogenesis, there is an accumulation of amyloid-β (Aβ) and the creation of intracellular neurofibrillary tangles (NFTs), which are composed of tau proteins [49,50]. Due to pathological events, the process of stabilization of microtubules by the tau process is disturbed, which further leads to tau aggregation and transformation into NFTs [51]. Pathological events leading to improper functioning of the tau protein may be hyperphosphorylated. This process becomes more frequent with cell aging. In the research of Alonso et al., it was proven that Hsp90, among co-chaperones, may regulate tau phosphorylation and dephosphorylation [52]. It is supposed to happen through the stabilization of tau kinases. Due to this process, it is possible to obtain a decrease in tau kinase activation by Hsp90 inhibition, resulting in the limitation of tau aggregation [53,54]. 

Some Hsp90 co-chaperones, like PP5, Cdc37, and CacyBP/SIP phosphatases, may dephosphorylate tau. Their level is higher with the aging of the cell [55,56]. Another Hsp90 co-chaperone, the CHIP protein, participates in the degradation of improper tau by starting ubiquitination. Loss of CHIP protein function leads to the pathological accumulation of tau [57]. Also, STI1/Hop damage was proven to lead to tauopathy. These findings suggest that losing the function of co-chaperones caused by aging is a cause of neurodegenerative conditions such as AD [58].

In most cases, the level of co-chaperones is limited, like when the level of FKBP52 co-chaperone is significantly decreased, which correlates with NTF aggregation or the level of FK506. However, the exception is FKBP51, whose amount is increased in AD patients [59,60,61]. 

Stress-inducible phosphoprotein 1 (STIP1) is one of many co-chaperones required for resilience to cellular stress. It migrates protein clients between Hsp70 and Hsp90. Additionally, STIP1 transmits signals through the cellular prion protein (PrPc). Under stress conditions, STIP1 concentrates in the nucleus, decreasing its extracellular level. In Alzheimer’s disease, when the concentration of soluble oligomers of amyloid-beta peptide (AβO) is high, it binds to PrPc, leading to cell death. STIP1 inhibits AβO and PrPc binding and activates neuroprotective signaling pathways using Ca^2+^ influx and PrPc. It remains unknown whether it could limit the toxicity of AβO in AD in vivo [62,63].

Another interesting chaperone is tumor necrosis factor receptor-associated protein 1 (TRAP1), which functions as an adaptive answer to counter cellular stresses contrary to maintaining housekeeping protein homeostasis [64]. This protein maintains equity between oxidative phosphorylation (OXPHOS) and aerobic glycolysis [65]. In AD, there is decreased activity of complex IV in the mitochondria, which causes oxidative damage. Thanks to TRAP 1, it is possible to inhibit complex IV activity. This chaperone cooperates with mitochondrial proto-oncogene tyrosine-protein kinase (c-Src) [66,67]. Due to these mechanisms, it is possible to change the metabolism of mitochondria from metabolic respiration to aerobic glycolysis, which limits the number of ROS produced. These molecules are produced in the highest amounts, especially in mitochondria [68,69]. Elevated oxidative stress and mitochondrial dysfunction are observed in the early stages of AD. Those events lead to protein aggregation [70]. The level of TRAP1 is elevated along with the increase in ROS production; however, under the influence of a substance such as iron chelator deferoxamine (DFO), the TRAP1 level is reduced, and the ROS amount is elevated [66]. What is more, TRAP1 regulates the permeability transition pore (PTP) [71]. This chaperone inhibits Cyclophilin D (CypD), a mitochondrial matrix protein, which regulates the formation of pores in the mitochondrial inner membrane [72]. TRAP1 prevents pore-opening, the same as Hsp90, due to blocking the conformational shift of CypD [73]. A decrease in CypD in the mitochondria prevents mitochondrial swelling caused by Aβ and Ca^2+^.

While TRAP1 is limited, the cytosol level of apoptogenic proteins such as Cyt C is increased, and an elevated level of caspase-3 activity leads to apoptosis. PTEN-induced putative kinase 1 (PINK1) phosphorylates TRAP1. Due to this process, TRAP1 is unable to inhibit Cyt C release by CypD [74].

Mitochondrial swelling caused by Aβ and Ca^2+^ is an important component in AD pathogenesis. Calcium dysregulation leads to hyperphosphorylation of tau and increased Aβ formation [75,76]. TRAP1 is also engaged in calcium regulation in the cell. Inhibition of this chaperone increases Ca^2+^ discharge in the mitochondria. An excessive amount of Ca^2+^ is finally released to the cytoplasm via the opened PTP [77,78]. 

Inhibition of TRAP1 also leads to a significant increase of glutathione (GSH) in the cell, which is compensation for the ROS increase. However, in AD patients, the level of GSH is decreased, which is an insufficient antioxidant amount [65,79].

Hsp90 is also involved in chronic microglial activation, which leads to neuronal damage due to the release of pro-inflammatory cytokines [80]. The role of microglia manifests in its dual activities—it may either protect or damage neurons. There are two types of activation, which differ by the type of released cytokines. The protective model is characterized by an increased amount of anti-inflammatory cytokines such as IL-10, IL-4, IL-13, and TGF-β and decreased levels of proinflammatory cytokines like IL-1β, TNF-α, reactive oxygen species (ROS), STAT3, IL-6, IL-12, and IL-23. It is responsible for tissue preparation and angiogenesis. The other type of activation is connected with neuron loss due to defense against damaged cells. In the other type, the level of proinflammatory factors is increased. [81,82]. The type of microglia changes during the disease, from the neuroprotective model to the one with a reduction in the number of neurons [80]. Furthermore, Aβ indirectly activates inflammatory systems, which leads to the progression of AD due to the persistent microglial activation and release of pro-inflammatory cytokines [83]. 

The structure of neurons in the healthy brain is maintained by cytoskeletal proteins. Tau is responsible for stabilizing the shape of the axon. Through the course of AD progression, there are various post-translational modifications of the tau protein. These changes result in a decrease in tau affinity for microtubules. It leads to microtubules changing their form so that they can easily aggregate. The integrity of the neurons is no longer sustained due to the depolymerization of microtubules. This leads to the degeneration of the neurons [41]. As a response to pathological tau, there are different cellular strategies activated to limit the aggregation, such as chaperones. Chaperones, in cooperation with co-chaperones, try to fold tau into its native form. Hsp70 and Hsp90 are involved in this process. However, during the progression of AD, those repair processes are no longer efficient. It fails to maintain tau in the correct form to prevent aggregation [84]. As a result, the neuronal cell dies. As these changes occur in an increasing number of cells, the disease spreads all over the brain [41] (Table 2).

## 7. Chaperones as Diagnostic Markers 

Chaperones may help diagnose many disorders, for example, tumors [94,95], heart diseases [96], rheumatologic disorders [97], and infectious or neurodegenerative disorders [98]. Hsp70 levels increase as a result of the accumulation of physical and chemical agents or stressors during life. When stress agents occur, there is a promotion of genes coding Hsp70, and increased production is a cell’s response [96]. The proteins created should enhance cell resistance to stressors and recovery from injury. However, these mechanisms became less efficient with age [98]. Serum Hsp90 levels may indicate impaired cognitive functions. Gezen-Ak et al. have observed significantly decreased levels of these molecules in serum compared to the control group [99]. It would help diagnose people with AD before serious clinical symptoms occur. It would allow them to conduct therapy earlier and delay the development of the disease. Hsp60 level may be used as a marker of early disease due to its increase in lymphocytes in AD patients in contrast to controls [100,101]. Those features may also be helpful in the prognosis of the disease and monitoring therapeutic effects, especially in chaperone-based treatment [102]. This is why this topic must be further investigated.

## 8. Targeting Chaperones in the Treatment of Alzheimer’s Disease—Possibilities 

Knowing the role of stress induced by the misfolding of Aβ and tau proteins in the pathogenesis of AD, the importance of chaperones has been evidenced as they regulate protein folding and activity and supervise the refolding and degradation of misfolded or aggregated proteins. The main role in neurodegenerative diseases is assigned to heat shock proteins (Hsps) [103]. 

Some studies found that extracellular Hsp60 release has a significant role in the production of pro-inflammatory factors and the overactivation of microglia, which leads to the progression of neurodegenerative diseases, including AD [104]. Therefore, inhibiting Hsp60 expression might be a possible strategy of treatment. Examples of substances that target Hsp60 are mizoribine, parazolopyrimidine EC3016, and epolactaene [103]. Mizorbine affects protein-folding activity and inhibits the detachment of Hsp10, which is a co-chaperonin, from the Hsp60/Hsp10 complex. It also decreases the ATPase activity of the Hsp60/Hsp10 complex [105,106]. EC3016 is known to block ATP binding and its hydrolysis, while epolactaene, a microbial metabolite that is isolated from Penicillium sp., and its derivative, epolactaene tertiary butyl ester (ETB), act through covalent binding to cysteine residue Cys442, both leading to Hsp60 inactivation and inhibition of its protein-folding function [103,107] (Table 3). Other grip points of Hsp60 and its inhibitors can be investigated, and their use can be translated to AD.

Much data shows that Hsp70’s increased level of expression inhibits Aβ aggregation, interferes with the APP secretory route, and degrades tau and Aβ oligomers through the proteasome system [103]. Compounds with the rhodacyanine skeleton bind different allosteric sites of Hsp70. Among those substances, MKT-077 and YM-01 both lead to a rapid reduction of tau levels by targeting Hsp70 and inhibiting the ATPase activity of the Hsp70/Hsp40 complex [108]. However, they show lower blood-brain barrier (BBB) penetration than YM-08, another molecule of similar effect [103]. Another group of molecules that act by inhibiting the ATPase function of Hsp70 are phenothiazines, including Methylene Blue and Azure C. This leads to a reduction of tau levels, both total and phosphorylated. They also stop tau accumulation and aggregation by interacting with its toxic oligomers [109,110,111]. However, they do not reverse the already existing neurofibrillary tangles [112]. There have been many molecules that modulate Hsp70 activity, and their possible use in therapy is still under research. Those are, among others, geranylgeranylacetone, celastrol, YC-1, or J-147 [103]. J147 showed a protective effect in AD cells and memory and recognition-promoting effects and may also induce nerve growth factors to reduce neuronal damage [113] (Table 4). Hsp70 is therefore a promising potential target for the treatment of AD and other tauopathies.

Hsp90 is the best-studied heat shock protein when it comes to its role in AD. Its functions include folding proteins, refolding, and degrading those that are denatured in conditions of stress. It can also inhibit amyloid accumulation and Aβ formation. Extracellular Hsp90 is responsible for the activation of phagocytes and the Toll-like receptor-4 pathway, which leads to Aβ degradation. It also regulates tau metabolism [103]. However, as tau is accumulating, Hsp90 function leads to its further aggregation, which is why Hsp90 inhibitors were found effective in reducing tau levels and decreasing Aβ toxicity [114]. We can divide those substances into two groups: N-terminal (geldanamycin and its analogs, purine scaffold) and C-terminal inhibitors. Geldanamycin is a natural antibiotic isolated from the Streptomyces genus, but due to its toxicity, it is not used. However, its analogues, for example, 17-AAG, were developed, and they tend to clear tau protein. Another group of Hsp90 is purine scaffolds, such as EC102 and PU24FCI. They could also potentially reduce tau levels. C-terminal inhibitors include celastrol, novobiocin, and its derivatives such as KU-32 and A-4, which also have protective effects in AD [103,114,115] (Table 5).

Hsp90 can be targeted more specifically through co-chaperones. Among others, ATPase homolog 1 (Aha1) increases the activity of Hsp90, leading to the production of aggregated tau. Therefore, inhibiting the Hsp90/Aha1 complex is a promising strategy for AD treatment. KU-177, a novobiocin-based inhibitor, reduced the amount of insoluble tau in transgenic mice. Another co-chaperone that is worth mentioning is Cdc37, as its higher levels correspond with higher levels of tau. Hsp90/Cdc37 complexes can be inhibited using celastrol and Withaferin A [103,114]. Celastrol has also been reported to disrupt the p23 function, which decreases tau stability. P23 is also inactivated by gedunin. Increased expression of CHIP, an E3 ubiquitin ligase, and another co-chaperone of Hsp90 leads to negative regulation of tau levels, so drugs of such mechanisms could be used in tauopathies. The Hsp90/PPID complex is responsible for axonal degeneration. Its inhibitor, cyclosporin A, could slow that process; however, it has many side effects. High levels of FKB51 and FKBP2 are known to interact with tau, leading to the production of tau oligomers. There is an inhibitor of the Hsp90/FKBP2/AR complex known as MJC13, but it needs further study to determine whether its use can be implicated in treating AD [114].

Another chaperone worth considering is clusterin, which plays a significant role in AD pathogenesis. It is involved in the formation of Aβ; it is also responsible for regulating inflammation, controlling apoptosis, and clearing pathological proteins. Thus, clusterin might be a promising field for scientists to develop new treatment strategies targeting this chaperone [27,116].

## 9. Targeting Chaperones in the Treatment of Alzheimer’s Disease—Limitations

Although the implementation of substances targeting chaperones in the treatment of AD seems like a promising field for scientists, it has its limitations and possible difficulties. The main reason is that most of the studies on substations mentioned in the previous paragraphs were performed in vitro. Thus, some difficulties in implicating these methods in humans can be expected. The main problems are the inability to simulate in vivo conditions in the assay or the lack of systems to specifically identify the chaperone’s involvement. That is why there might be some false-positive or false-negative drugs. The creation of a novel assay that can demonstrate both the antiaggregating effect of a molecule as well as its action through chaperones is needed to reduce their number and enable scientists to conduct research focusing mainly on the safety of these molecules [117].

Another obstacle to translating the mentioned substances from preclinical to human clinical trials is their possible side effects and poor pharmacokinetic properties. For example, the use of geldanamycin, which has a high affinity for the ATPase domain of Hsp90, is limited due to its toxicity and pharmacokinetics, as the blood-brain barrier shows low permeability for this substance. Similarly, the use of 17-AAG, which is proven to be safe in cancer treatment, in AD remains restricted due to poor blood-brain barrier permeability. Moreover, due to its toxicity, the clinical application of 17-AAG is questionable, despite promising results in disease models. The same applies to celastrol, which, despite its effectiveness, is characterized by cytotoxicity, so further research should be carried out on the safety of its use in neurodegenerative diseases [118]. The development of Hsp inhibitors that induce heat-shock responses without cytotoxicity and present better properties when it comes to, for example, penetrating the blood-brain barrier is needed [119]. An important aspect is also selecting an effective but safe dosage and designing novel synthetic or semi-synthetic chaperone inhibitors that are more selective and present lower off-target toxicity than the natural ones [120,121].

Although chaperones seem to be ideal candidates for novel inhibitory targets, their molecular mechanisms associated with neuroprotection are still poorly elucidated [119,122]. Multiple fields need studying to fully understand the potential of using drugs targeting chaperones in AD, such as the mode of action of Hsp inhibitors, biochemical pathways involving Hsps in the course of AD, interactions between client/Hsps protein-protein, and selective targeting of constitutive compared to stress-induced Hsps [103]. Some studies also show that targeting chaperones might be effective only in the early stages of AD, as the formation of fibrils is not reversed by the action of the Hsp proteins, whereas oligomers seem susceptible to manipulation. An important question is also whether oligomers of Aβ are present in the same subcellular space as chaperones and if direct physical interaction between those agents is required. Aβ fibrils are located in the extracellular regions, whereas chaperones are expressed in the intracellular regions. However, the exact location of Aβ oligomer formation is still unclear and requires further research [123].

## 10. Final Remarks and Conclusions

A precise understanding of AD pathogenesis allows for the invention of new therapy methods. This is the hope for limiting the future expenses of healthcare for AD patients and improving the lives of people suffering from these conditions by delaying the breakthrough of the disease or reducing the severity of the symptoms. An increasing number of studies indicate that improperly functioning chaperones and co-chaperones are essential factors in the pathogenesis of AD. Proteins like Hsp90 or Hsp70 play an important role in maintaining cell homeostasis. However, disturbances in their functioning due to stress factors may result in Aβ and tau aggregation. Exclusively, heat shock protein (Hsp) chaperones are critical regulators of neurodegenerative diseases. Hsp70 and Hsp90 may prevent the early stages of amyloid aggregation [124]. Damage to regulation by other molecules like co-chaperones, for example, PP5, Cdc37, CacyBP/SIP, the CHIP protein, and STI1/Hop, is also crucial for AD pathogenesis due to the disturbance of molecular pathways of cell homeostasis regulation. Hsp60 and Hsp70 activity may be regulated by some substances that may be used in creating possible therapy methods. Some substances may either reduce tau levels or inhibit tau accumulation and aggregation, and some may also neuroprotect from Aβ toxicity.

Many aspects of the discussed issues certainly need further investigation; the present revelations allow us to make optimistic predictions for the next few years. Precise discovery of molecular impacts is undoubtedly needed to improve disturbances leading to AD.

## Figures and Tables

**Figure 1 ijms-25-03401-f001:**
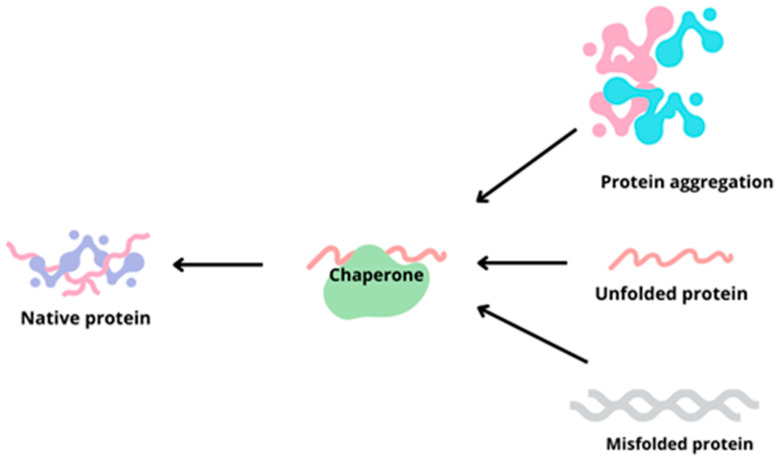
The major chaperone function is based on the information from [9,10,11]. Chaperones can create and recreate the proper native protein from abnormal and incorrect protein structures (aggregated, unfolded, or misfolded ones).

**Figure 2 ijms-25-03401-f002:**
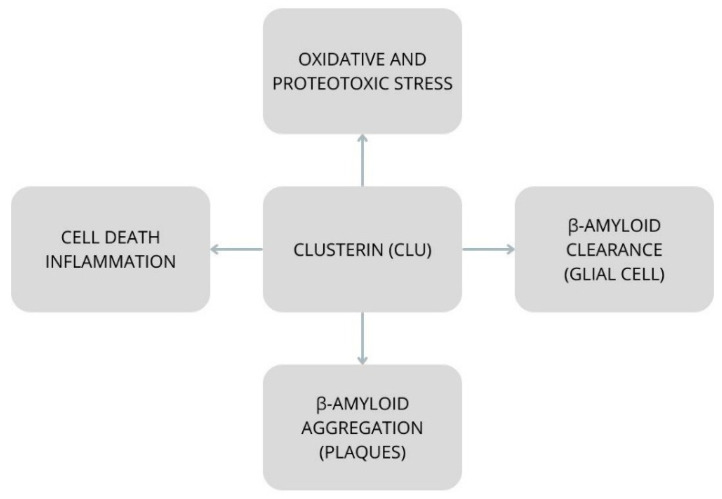
Functions of clusterin based on [27,29,32,37,38].

**Figure 3 ijms-25-03401-f003:**
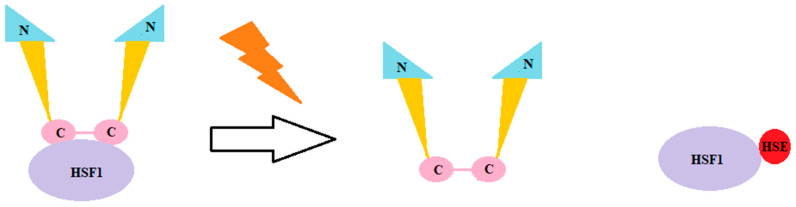
Hsp90 in stress conditions. Hsp90 consists of three parts: the N-terminal ATP-binding domain, the center domain, and the C-terminal dimerization domain. Hsp90 may be regulated by interactions with co-chaperones, ATP, or heat shock factor (HSF1) [45,46]. In cases of stress conditions, HSF1 disconnects from Hsp90 and connects with heat shock factor elements (HSEs) on the sequence of the Hsp90 encoding gene [47].

**Table 1 ijms-25-03401-t001:** Characteristics of Hsps [15].

Chaperone:	Hsp60	Hsp70	Hsp90	Hsp100	Shop
Function:	Segregation of unfolded polypeptides; promotion of unfolding for misfolded polypeptides (by active and passive mechanisms)	The unfolding of misfolded polypeptides, translocation of unfolded polyproteins, dissociation of protein complexes	Modification of specific proteins and transcription factors.	Dissociation, refolding, un-aggregation.	Protection of proteins from irreversible aggregation
Localization:	Cytoplasm Mitochondria	Cytoplasm Endoplasmic Reticulum Mitochondria Nucleus	Cytoplasm Endoplasmic Reticulum Mitochondria Nucleus	Cytoplasm Nucleus	Cytoplasm Mitochondria
ATP-binding activity	+	+	+	+	-

**Table 2 ijms-25-03401-t002:** Chaperones and co-chaperone characteristics in the context of AD.

Chaperone and Co-Chaperones	Level in AD	Function in Cell
Hsp60	increased [85]	Inhibiting Aβ amyloid aggregation by closing molecular pathways leading to peptide fibrillogenesis [86]
Hsp70	increased [85]	Protecting neurons from intracellular accumulation of Aβ through promoting the clearance of Aβ [87]
Hsp90	increased [42]	Regulating tau phosphorylation [42]
p23	decreased [88]	Facilitating the adenosine triphosphate-driven cycle of Hsp90 binding to client proteins [88]
PP5	decreased [89]	Dephosphorylation of tau [89]
STI1/Hop	increased [90]	Clearance of tau [91]
Stg1	decreased [42]	Clearance of tau [42]
CHIP	decreased [92]	Tau ubiquitination [42]
FKBP51	increased [93]	Promoting, in coordination with Hsp90, the accumulation of non-ubiquitinated tau in the presence of a proteasome inhibitor [93]

**Table 3 ijms-25-03401-t003:** Examples of substances targeting Hsp60 and their effects.

Substance	Mechanism	Effect
Mizorbine	- Inhibiting the detachment of Hsp10 - decreasing the ATPase activity of Hsp60/Hsp10 [105,106]	Inhibition of protein folding function [105,106]
Parazolopyrimidine EC3016	- blocking ATP binding and its hydrolysis [103,107]	Hsp60 inactivation, inhibition of protein folding function [103,107]
Epolactaene	- binding to Cys442 [103,107]	Hsp60 inactivation, inhibition of protein folding function [103,107]

**Table 4 ijms-25-03401-t004:** Examples of substances targeting Hsp70 and their effects.

Group of Molecules	Representatives	Mechanism	Effect
Compounds with rhodacyanine skelton	- MKT-077 - YM-01 - YM-08	Inhibiting ATPase activity of Hsp70/Hsp40 complex [103,108]	Rapid reduction of tau levels [103,108]
Phenothiazines	- Methylene Blue - Azure C	- Inhibiting ATPase function of Hsp70 - Interacting with toxic tau oligomers [108,109,110,111]	- Reduction of tau levels - Inhibiting tau accumulation and aggregation [108,109,110,111]

**Table 5 ijms-25-03401-t005:** Examples of substances targeting Hsp90 and their effects.

Group	Representatives	Effect
N-terminal inhibitors	Geldanamycin 17-AAG EC102 PU24FCI	Clearing tau protein, reducing tau levels [103,114,115]
C-terminal inhibitors	celastrol novobiocin KU-32 A-4	Neuroprotection from Aβ toxicity [103,114,115]

## Data Availability

No new data were created.

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
