# Peer review of "Chaperones—A New Class of Potential Therapeutic Targets in Alzheimer’s Disease"

_ijms, 2024, doi:10.3390/ijms25063401_

Round 1

Reviewer 1 Report

Comments and Suggestions for Authors

This is an extensive review highlighting the roles of molecular chaperones in neurotoxicity in AD.

Major concern:

The authors are required to amend the content of the current manuscript based on the following two issues. A systematic review would be able to justify your up-to-date findings.

1. The main title of the manuscript is almost similar to a published review in 2009:

Koren, J., 3rd, Jinwal, U. K., Lee, D. C., Jones, J. R., Shults, C. L., Johnson, A. G., Anderson, L. J., & Dickey, C. A. (2009). Chaperone signalling complexes in Alzheimer's disease. Journal of cellular and molecular medicine13(4), 619–630. https://doi.org/10.1111/j.1582-4934.2008.00557.x

2. There could be some overlapping of information between the current review and recently published review 

Wankhede, N. L., Kale, M. B., Upaganlawar, A. B., Taksande, B. G., Umekar, M. J., Behl, T., Abdellatif, A. A. H., Bhaskaran, P. M., Dachani, S. R., Sehgal, A., Singh, S., Sharma, N., Makeen, H. A., Albratty, M., Dailah, H. G., Bhatia, S., Al-Harrasi, A., & Bungau, S. (2022). Involvement of molecular chaperone in protein-misfolding brain diseases. Biomedicine & pharmacotherapy = Biomedecine & pharmacotherapie147, 112647. https://doi.org/10.1016/j.biopha.2022.112647

Subsection 3 is too broad. I suggest : Types of chaperones and Regulation of chaperones in AD.

It is unclear for Section 4 whether Clusterin is also chaperone? A precise heading would be ideal.

To include a section of Limitations and Future Prospect.

Comments on the Quality of English Language

-

Author Response

Thank you very much for taking the time to review this manuscript and for highlighting topics that could be improved. 

Response to comments and suggestions: 

1. "The main title of the manuscript is almost similar to a published review in 2009" We agree with the comment and therefore we changed the title to: 

"Chaperones - new class of potential therapeutic targets in Alzheimers disease"

2. Subsection 3 is too broad - we divided this paragraph into two parts and precised some information about chaperonopathies

3. It is unclear for Section 4 whether Clusterin is also chaperone - we tried to clarify and analise this matter in the section 4

4. To include a section of Limitations and Future Prospect - such section is included in the new version of our manuscript 

We also edited the language used in the manuscript and corrected typos. 

Yours sincerely,

Authors

Reviewer 2 Report

Comments and Suggestions for Authors

The Review «Chaperone signaling in Alzheimer’s Disease» describes correlations between impaired functioning of chaperones and co-chaperones in Alzheimer disease (AD) pathogenesis. Recent discoveries point to the significant impact of the dysfunction of chaperones and co-chaperons on AD pathogenesis. Authors characterize Alzheimer’s Disease - neurodegenerative disease and the most common cause of dementia. It is caused by the accumulation of the amyloid-beta (Aβ) outside of neurons. New therapies of AD involving monoclonal antibodies also target Aβ, e.q., aducanumab and gantenerumab. Next Chapter is about Chaperons - they are a group of specific proteins capable not only of folding the unfolded polypeptides but also of repairing misfolded ones. Additionally, they are capable of preventing protein aggregation as well as directing terminal proteins for proteolytic degradation. The imbalance of Chaperone (chaperoning) system (CS) is observed as a formation and accumulation of pathological inclusions, α-synuclein in Parkinson’s disease or huntingtin in Huntington’s disease, as well as the extracellular β-amyloid plaques in Alzheimer’s disease. There is characteristic of chaperon's groups - Hsp40s, Hsp60s, Hsp70s, Hsp90s, Hsp100s. The other Chapter is about Clusterin (CLU), or APOJ - a multifunctional glycoprotein implicated in several physiological and pathological states, including Alzheimer's disease (AD). Elevated clusterin levels in cerebrospinal fluid, brain and plasma have been observed in AD patients. Clusterin’s ability to interact and bind to Aβ appears to alter aggregation and promote Aβ clearance, suggesting a neuroprotective role. Authors cited several studies about involvement of clusterin in AD progress. There is a separate Chapter about «Hsp90, Hsp70, co-chaperones» - more detailed describing of some Hsp90 co-chaperones, like PP5, Cdc37, and CacyBP/SIP, involvement in AD pathogenesis. Stress-inducible phosphoprotein 1 (STIP1) is one of many co-chaperones being required for resilience to cellular stress. It migrates protein clients between Hsp70 and  Hsp90. Tumor necrosis factor receptor-associated protein 1 (TRAP1) which functions as an adaptive answer to counter cellular stresses contrary to maintaining housekeeping protein homeostasis. This chaperone inhibits Cyclophilin D (CypD), a mitochondrial matrix protein, which regulates formation of pores in the mitochondrial inner membrane. There is a Chapter about Targeting chaperones in treatment of Alzheimer Disease. For example Mizorbine affects protein-folding activity and inhibits the detachment of Hsp10, which is a co-chaperonin, from the Hsp60/Hsp10 complex; Parazolopyrimidine EC3016 blockS ATP binding and its hydrolysis that leads to Hsp60 inactivation, inhibition of protein folding function; compounds with rhodacyanine skeleton (MKT-077 and YM-01) bind different allosteric sites of Hsp70 that lead to inhibiting the ATPase activity of Hsp70/Hsp40 complex, etc. In the end 

The issuie of Review is relevant for the field. But there are similar reviews published recently: Antonella Marino Gammazza - Alzheimer's Disease and Molecular Chaperones: Current Knowledge and the Future of Chaperonotherapy, 2016; Jessica Tittelmeier - Molecular Chaperones: A Double-Edged Sword in Neurodegenerative Diseases, 2020; Nitu L. Wankhede - Involvement of molecular chaperone in protein-misfolding brain diseases, 2022. Of course, they are just similar, and current Review has its own unique, but it should be taken into account. The cited references contain recent publications (within the last 5 years) but it could be improved. There is not excessive number of self-citations.

As for the structure of presentation - Figures (schemes) is not described enough and do not understandable. It should be described in more detail in the text and in the captions.

Author Response

Thank you very much for taking the time to review this manuscript. Please find the detailed responses below.

Response to Comments and Suggestions for Authors:

  1. The cited references contain recent publications (within the last 5 years) but it could be improved. - in newly added parts we tried to cite as many recent publications (from 2022, 2023 and 2024) as possible.
  2.  Figures (schemes) is not described enough and do not understandable. It should be described in more detail in the text and in the captions. - in the current version of the manuscript we improved the captions of figures to make them more understandable

Yours sincerely,

Authors

Reviewer 3 Report

Comments and Suggestions for Authors

This review article discusses the role of chaperone signaling in the pathogenesis of AD. Chaperones and co-chaperones are critical for regulating protein folding, refolding misfolded proteins, and degrading aggregated proteins to maintain cellular homeostasis. However, in AD, the function of key chaperones like Hsp90 and Hsp70 becomes impaired, leading to the accumulation and aggregation of amyloid-beta (Aβ) and tau proteins. Dysregulation of co-chaperones such as PP5, Cdc37, CacyBP/SIP, CHIP, FKBP52, and STIP1 also contributes to AD pathology by disrupting the regulation of Aβ and tau. Targeting chaperones and co-chaperones presents potential therapeutic strategies for AD. Inhibiting Hsp60 expression may reduce neuroinflammation, while compounds that target Hsp70, such as rhodacyanine and phenothiazine derivatives, can reduce tau levels and aggregation. Hsp90 inhibitors, including geldanamycin analogues and C-terminal inhibitors like novobiocin, have shown promise in reducing Aβ toxicity and tau pathology. Additionally, modulating specific Hsp90 co-chaperones like Aha1, Cdc37, and CHIP could further mitigate AD-related protein aggregation. The chaperone clusterin is also implicated in AD and represents another potential therapeutic target.  

  1. The review may not adequately address conflicting evidence or alternative hypotheses regarding the role of chaperones in AD pathology. Including a more balanced discussion of the limitations and inconsistencies in the current research would improve the article's objectivity.
  2. While the article mentions some potential therapeutic strategies targeting chaperones, it does not thoroughly address the possible side effects of these interventions. A more comprehensive discussion of the risks and limitations of targeting chaperones would strengthen the review.
  3. The article discusses various chaperones and co-chaperones individually but does not provide a clear picture of how they interact and work together in the context of AD. Exploring the interplay and crosstalk between different chaperones would provide a more comprehensive understanding of their roles in AD pathology.
  4. The review focuses primarily on the downstream effects of chaperone dysregulation in AD but does not extensively discuss the upstream factors that may contribute to chaperone dysfunction. Investigating the potential role of genetic, environmental, and age-related factors in chaperone dysregulation could provide a more complete picture.
  5. While the article highlights potential therapeutic strategies, it does not sufficiently address the translational challenges in moving these interventions from preclinical studies to human clinical trials. Discussing the potential obstacles and strategies to overcome them would make the review more relevant to translational research efforts.
  6. The article could benefit from exploring the potential of chaperones and co-chaperones as biomarkers for AD diagnosis, prognosis, and treatment response monitoring. Discussing the current state of research on chaperone-based biomarkers and their potential clinical utility would strengthen the review.
  7. The review does not clearly address how chaperone dysfunction may evolve throughout the course of AD progression. Exploring the temporal dynamics of chaperone dysregulation and its relationship to specific stages of AD could provide valuable insights for targeted interventions.
  8. While the article focuses on potential pharmacological interventions targeting chaperones, it could benefit from discussing non-pharmacological approaches, such as lifestyle modifications or cognitive training, that may influence chaperone function and AD risk.
  9. The review would be more impactful if it concluded with a clear set of recommendations for future research directions based on the identified gaps and limitations in the current understanding of chaperone signaling in AD. Providing a roadmap for future studies would help guide the field toward addressing the most pressing questions and advancing translational efforts.

Author Response

Thank you very much for taking the time to review this manuscript. Please find the responses below.

After analizing all of the suggestions we added a paragraph that concentrates on possible limitations and difficulties in translating the approach of targetting chaperones in AD treatment from preclinical to clinical trials. We also mentioned the possible side affects of chaperones' inhibitors.

We also added a paragraph about possible use of chaperones as diagnositc markers.

We enhanced the paragraph considering chaperones and their functions with information about the dynamics of chaperone dysregulation and its relationship to specific stages of AD.

Yours sincerely,

Authors

Reviewer 4 Report

Comments and Suggestions for Authors

In this review, the authors aim to highlight the importance of chaperones in Alzheimer's disease, in particular - Hsp 60, 70, 90 & clusterin, and discuss the various inhibitors that can be potentially used to alleviate Abeta and tau-mediated toxicity. Although the manuscript discusses various aspects of chaperone dysregulation and therapeutic potential, it could benefit from better information in the Figures, detailed tables, and discussion on the caveats of chaperone as a therapeutic target.

Comments on the manuscript:

Abstract: This can be modified to better define the aim and significance of this review. Chaperones and co-chaperones which are discussed in detail in the manuscript can be defined and introduced better in the abstract. For example, PP5, Cdc37, 16 CacyBP/SIPTRAP1 are Hsp 90 co-chaperones. 

Line 32: In 2050

Line 36: The significance of this review needs to be clearly defined in the introduction. Of all the dysfunctions in AD pathogenesis, why is chaperone signaling an important target is not clearly stated in this section.

Sub-heading: Alzheimer's disease

Various cellular mechanisms that are dysregulated in AD need to be discussed first.

Figure 1: Can be modified to include the following details:

I. the effects of chaperones in 1) preventing aggregation 2) disaggregation 3) Sorting misfolded proteins to various cellular compartments for degradation/processing. 

II. Various chaperones in Table 1

III. Chaperoning system- Chaperones and Co-Chaperones can be included as figure.

Typo: Fig 1: Chaperon

Line 95: Modify the chaperoning system line 

Line 100: Define the term Chaperonopathy here.

Line 106-108: Not needed

Line 123: Hsp 70

Table 1 can be modified to include

I. Chaperones and its co-chaperones

II. ATP binding activity

III. Functions of the chaperone, co-chaperone.

IV. Localization of the cellular compartment

Line 137: Define clusterin from Line 146 here 

Figure 2 caption can be modified as Functions of clusterin

Line 182: Conformation

Line 184: Modify line "main proteins stored in AD"

Line 187: "stress factors"

Line 188: "chaperones"

Line 193: It's not clear what are referred to as the listed proteins

Figure 3: Very vague. This figure needs to include the binding effect of HSE- downstream functions, and how it affects AD pathogenesis.

Line 221: "Loss of function" of which protein?

New Table: A new table that includes Chaperones/Co-chaperones, potential pathogenesis reported in AD, and increase or decrease in levels reported in AD patients will be very useful to see the chaperone dysregulation in nutshell.

Line 232: under stress conditions

Line 255: TRAP1 repetitive information

Line 267: "significant growth" --> significant increase

Line 271: "Dual role" repetitive

Table 2,3,4 : Modify to include specific references directly along with the table

Line 337: Hsp 90 not Hsp 70

New figure 4: A new figure with the chaperones/co-chaperones and their inhibitors tabulated in 2,3 &4 and observed effect would be a good visual representation

Overall in the manuscript, various inhibitors of chaperone are discussed. But the role of overexpressed chaperones, engineered chaperones, chaperones in nucleation/elongation of fibrils, seeding, and propagation by chaperones are not discussed, this would help in getting the overall roles of chaperones.

Discussion on the potential drawbacks of using chaperone inhibitors is missing: 

1) Inhibiting a particular chaperone --> Compensatory upregulation of other chaperones.

2) Chaperones as disaggregase --> At later stages, could inhibitors be effective treatment if they do not act on pre-formed fibrils.

3) How does altering chaperone levels in a continuously changing pathogenic environment in the cells affect other proteins.

Line 374: Vague interpretation

Comments on the Quality of English Language

The quality of language and the flow of the manuscript has scope for improvement for better readability. Some typos need to be corrected!

Author Response

Thank you very much for taking the time to review this manuscript. Please find the responses below.

After studying all of th suggestions, we made significant changes to our manuscript. We modified the abstract and introduction to better clarify the reason why is chaperone signaling an important target in AD treatment. We also modified the description of Figures to make them more understandable. We applied suggested changes to Tables and we included a new table that describes Chaperones/Co-chaperones and their potential pathogenesis reported in AD. A new paragraph considering the limitations and difficulties in using chaperone inhibitors in AD treatment was also added. We corected the typos and edited the lenguage used in the manuscript.

Your sincerely, 

Authors 

Round 2

Reviewer 1 Report

Comments and Suggestions for Authors

None

Reviewer 2 Report

Comments and Suggestions for Authors

The current version of the Reveiw was improved enough. The Figures were described in detail and the References were expanded.

Reviewer 4 Report

Comments and Suggestions for Authors

Authors have duly addressed majority of the comments.

Comments on the Quality of English Language

Many typos were corrected and better, clear sentences are used.